# Non-Parenchymal Cells and the Extracellular Matrix in Hepatocellular Carcinoma in Non-Alcoholic Fatty Liver Disease

**DOI:** 10.3390/cancers15041308

**Published:** 2023-02-18

**Authors:** Koen C. van Son, Lars Verschuren, Roeland Hanemaaijer, Helen Reeves, R. Bart Takkenberg, Joost P. H. Drenth, Maarten E. Tushuizen, Adriaan G. Holleboom

**Affiliations:** 1Department of Vascular and Internal Medicine, Amsterdam University Medical Center, 1105 AZ Amsterdam, The Netherlands; 2Department of Gastroenterology and Hepatology, Radboud University Medical Center, 6525 GA Nijmegen, The Netherlands; 3Department of Metabolic Health Research, Netherlands Organization for Applied Scientific Research, 2333 BE Leiden, The Netherlands; 4Newcastle University Translational and Clinical Research Institute, Newcastle upon Tyne NE2 4HH, UK; 5Department of Gastroenterology and Hepatology, Amsterdam University Medical Center, 1105 AZ Amsterdam, The Netherlands; 6Department of Gastroenterology and Hepatology, Leiden University Medical Center, 2333 ZA Leiden, The Netherlands

**Keywords:** non-alcoholic fatty liver disease, non-alcoholic steatohepatitis, hepatocellular carcinoma, hepatocarcinogenesis, non-parenchymal cells, hepatic stellate cells, macrophages, liver sinusoidal endothelial cells, immune surveillance

## Abstract

**Simple Summary:**

The incidence of hepatocellular carcinoma (HCC) in patients with non-alcoholic fatty liver disease (NAFLD) has increased in recent years. Compared to HCC caused by other chronic liver diseases, NAFLD-related HCC is often detected later, because it more commonly arises before cirrhosis has occurred. Because of this late diagnosis, NAFLD-related HCC is often more advanced at time of diagnosis, resulting in fewer curative treatment options. Most research in the pathogenesis of HCC has focused on the disease processes in hepatocytes, the most abundant type of liver cells. However, other cell types, such as cells of the immune system and cells that regulate connective tissue formation, also play an important role in the development of NAFLD-related HCC, both by contributing to the development of HCC itself and by interfering with the immune system’s ability to attack cancer cells. In this paper, we review the role of different cell types in the development of NAFLD-related HCC.

**Abstract:**

Hepatocellular carcinoma (HCC) in the setting of non-alcoholic fatty liver disease (NAFLD)-related cirrhosis and even in the pre-cirrhotic state is increasing in incidence. NAFLD-related HCC has a poor clinical outcome as it is often advanced at diagnosis due to late diagnosis and systemic treatment response is poor due to reduced immune surveillance. Much of the focus of molecular research has been on the pathological changes in hepatocytes; however, immune cells, hepatic stellate cells, liver sinusoidal endothelial cells and the extracellular matrix may play important roles in the pathogenesis of NAFLD-related HCC as well. Here, we review the role of non-parenchymal cells in the liver in the pathogenesis of HCC in the context of NAFLD-NASH, with a particular focus on the innate and the adaptive immune system, fibrogenesis and angiogenesis. We review the key roles of macrophages, hepatic stellate cells (HSCs), T cells, natural killer (NK) cells, NKT cells and liver sinusoidal endothelial cells (LSECs) and the role of the extracellular matrix in hepatocarcinogenesis within the steatotic milieu.

## 1. Introduction

Hepatocellular carcinoma (HCC) represents the sixth most common malignancy worldwide and the fourth most common cause of cancer-related mortality [1]. It occurs predominantly in patients with advanced stages of chronic liver disease [2,3]. Non-alcoholic fatty liver disease (NAFLD) is a rapidly growing cause of chronic liver disease which is closely associated with the epidemic of obesity, metabolic syndrome and type 2 diabetes mellitus [4,5,6]. In the global population, NAFLD, which is defined as the accumulation of intracellular fat in >5% of hepatocytes, reaches prevalence rates of over 25% [5] while in patients with obesity and/or type 2 diabetes mellitus prevalence rises to 60–80% [6,7]. The spectrum of NAFLD ranges from isolated steatosis, characterized by lipid accumulation in hepatocytes, to non-alcoholic steatohepatitis (NASH) with the addition of hepatic inflammation, and NASH-related fibrosis, cirrhosis and HCC [7,8]. Progression along the NAFLD disease spectrum often goes unnoticed until advanced stages of fibrosis or even cirrhosis or HCC occur. This becomes ever more clinically relevant since patients are living longer with more severe obesity and type 2 diabetes mellitus, driving the occurrence of advanced fibrotic stages of disease, which are in turn associated with increased liver-related mortality and all-cause mortality [9,10].

Currently, there are no amply sized prospective cohort studies that allow for the calculation of risk of developing HCC in patients with NAFLD. Data on the incidence of HCC in this setting are conflicting, but most reports agree that the incidence rate of HCC is generally lower in the setting of NAFLD compared to that of other common etiologies such as viral hepatitis [11,12]. Yet, in a U.S. population-based study involving 4406 patients with HCC, NAFLD was found to be the underlying disease in 59% of cases [13]. Recent estimates predict that the incidence of NAFLD-related HCC will increase dramatically by 2030, with expected increases of 117% and 122% in France and the United States, respectively [14]. In the United States, NAFLD is now the fastest growing underlying cause of HCC in liver transplant recipients and transplantation candidates on the waiting list [15].

The pathophysiology of NAFLD-NASH is complex, and has been extensively reviewed [16,17,18]. Briefly, obesity and insulin resistance are key drivers, the latter triggering an increased flux of circulating free fatty acids (FFAs) from insulin-resistant peripheral adipose tissue to the liver [19,20,21]. These FFAs are stored as triglycerides in lipid droplets, reducing hepatic insulin sensitivity and consequently increasing hepatic gluconeogenesis, which results in hyperglycemia and intrahepatic conversion of glucose to FFAs, further compounding intrahepatic fat accumulation [22]. Meanwhile, high plasma insulin levels increase de novo lipogenesis, producing even more triglycerides and further enhancing hepatic gluconeogenesis. When hepatic mechanisms for storage, secretion or beta-oxidation fall short, lipotoxicity occurs, causing mitochondrial dysfunction, resulting in the formation of reactive oxygen species (oxidative stress), necro-inflammation of hepatocytes and an influx of monocytes and lymphocytes [23,24,25,26,27,28]. Within this lipotoxic and inflammatory milieu and compounded by the inhibition of endothelial nitric oxide synthase (eNOS) mediated by liver sinusoidal endothelial cells (LSECs) [29], quiescent hepatic stellate cells (HSCs) differentiate into myofibroblasts which secrete a fibrotic matrix, rich in type I collagen [30,31] in a process of damage and repair which is maladaptive in the chronic state of NASH and thus leads to progressive fibrosis [16].

During this process of NASH and fibrogenesis, regenerative repair pathways are induced such as the Hedgehog (Hh) signaling pathway, which can ultimately induce hepatocarcinogenesis [18,32]. Most research into the development and subsequent progression of NAFLD-related HCC has focused on hepatocytes, the cell type of origin in HCC [33,34]. Yet, of note, the tumor microenvironment (TME) also consists of stromal cells, endothelial cells, immune cells, cytokines and extracellular matrix (ECM), and these may all play a key role in the initiation and progression of HCC [18,35]. Here, we review the roles of non-parenchymal cells, including macrophages, lymphocytes, LSECs and HSCs, in the initiation and progression of NAFLD-related HCC. Furthermore, we will assess the role of the ECM and matrix stiffness as drivers of NAFLD-related HCC. We will consider both the processes that underlie the initiation of NAFLD-related HCC and the processes that underlie its eventual progression.

## 2. Hepatocyte Injury in HCC Pathogenesis: A Brief Summary

First, we will briefly assess the pathogenesis underlying the initiation of NAFLD-related HCC, which has been extensively reviewed elsewhere [18,32,36]. Reactive oxygen species (ROS) are generated in response to elevated mitochondrial fatty acid oxidation and inadequate mitochondrial respiratory chain activity [32,37,38]. Moreover, when NAFLD progresses, macrophages and apoptotic hepatocytes are key contributors to ROS generation [37,39]. ROS can cause either point mutations or larger lesions in the genome, thus driving genomic damage and genetic instability [40,41]. This pro-carcinogenic effect is exacerbated as oxidative stress stimulates a DNA-damage response that results in error-prone DNA repair, further aiding genomic instability [18]. Additionally, ROS stimulates hepatocyte survival by stimulating the nuclear factor kappa-light-chain-enhancer of activated B cell (NF-κB) signaling and increases the release of pro-inflammatory cytokines, including tumor necrosis factor (TNF)-α and interleukin (IL)-6 [42,43,44,45]. This creates a self-propelling process where increased inflammation and oxidative stress exacerbate genomic instability [32,46,47]. 

In the setting of NAFLD, impaired autophagy further adds to oxidative stress in the liver [32]. Autophagy is a process aimed at the lysosomal degradation of damaged cell components [48,49]. As such, autophagy has antitumorigenic properties [50]. To counter this diminished autophagy and the rise in oxidative stress, hepatocytes induce the expression of the KEAP1-NRF2 pathway which has antioxidant properties [51]. However, this also allows for the expression of pro-survival genes and protects HCC-initiating cells from oxidative stress-induced death, thus contributing to hepatocarcinogenesis [32,50,52,53]. 

Hepatocyte apoptosis is significantly increased in NASH and constitutes a dual, yet opposing, role in the initiation of HCC [54,55,56,57]. On the one hand, it constitutes a protective mechanism by eliminating damaged hepatocytes, while on the other hand it increases liver regeneration and subsequent DNA replication stress [56]. This can ultimately lead to additional DNA damage and added genomic instability. Additionally, hepatocyte apoptosis adds to the pro-inflammatory environment and the recruitment of monocytes into the liver [58,59,60] and the release of hepatocyte apoptotic bodies can induce the activation of macrophages and HSCs [48,61]. 

Insulin resistance, one of the hallmarks of NAFLD [16,18], increases the secretion of insulin and insulin-like growth factor (IGF)-1. Binding of insulin or IGF-1 to their respective receptors triggers a signaling cascade that results in the activation of the downstream phosphoinositide-3 kinase (PI3K) and mitogen-activated protein kinase (MAPK) pathways [46]. These pathways play a significant role in hepatocarcinogenesis by the induction of hepatocyte proliferation and inhibition of apoptosis [46]. Pro-inflammatory cytokines, including TNF-α and IL-6, drive hepatocyte proliferation by activating stress-related signaling pathways, including c-Jun N-terminal kinase (JNK) and NF-κB signaling pathways, and signal transducer and activator of transcription (STAT)3 and extracellular signal-regulated kinase (ERK) signaling pathways, respectively. Furthermore, the Hh signaling pathway is activated in NASH because ballooned hepatocytes produce Hh molecules that are released upon hepatocyte injury [49,62]. Overstimulation of Hh signaling results in dysregulated cellular repair and subsequent malignant transformation [63]. 

In the next sections, we will assess the role of immune cells, LSECs and HSCs, in the initiation and progression of NAFLD-related HCC. 

## 3. Inflammatory Pathways in the Pathogenesis of NAFLD and NAFLD-Related HCC

The liver has complex immunological functions, as it acts as the physiological connection between gut-derived molecules and the systemic circulation. Liver macrophages consists of Kupffer cells (KCs), which are self-renewing, liver resident cells that serve as sentinels for liver homeostasis, and monocytes, which are not liver-resident but infiltrate the liver from the peripheral blood and bone marrow following liver injury [64]. As an essential part of liver homeostasis, KCs are immunotolerogenic in nature, so as to avoid the induction of immunity against harmless substances such as gut-derived nutrients [64,65,66]. However, upon activation, liver KCs lose this immunotolerogenic nature, and subsequently secrete growth factors, pro-inflammatory cytokines, chemokines, such as chemokine ligand CCL2, and ROS [49,67,68]. CCL2, also referred to as monocyte chemoattractant protein 1 (MCP1), triggers chemotaxis, prompting monocytes derived from the peripheral blood or bone marrow to infiltrate into the liver and to differentiate into macrophages [69,70]. This pro-inflammatory environment is enhanced by the release of TNF-α and IL-6 locally in the liver, but is also caused by systemic chronic inflammation processes derived from metabolic tissues such as adipose tissue resulting from insulin resistance [71], and by high leptin but low adiponectin levels, which are typically found in people who are overweight [71,72]. In all, this triggering of the innate immune response marks the transition of simple steatosis to actual steatohepatitis [68,73] and plays a key part in the initiation of NAFLD-related HCC. 

### 3.1. M1-Type Macrophages Drive Inflammation and Subsequent Hepatocarcinogenesis

Generally speaking, macrophages that encourage inflammation are called M1 macrophages, whereas those that decrease inflammation and encourage tissue repair are called M2 macrophages [26,68,74]. Yet, while anti-carcinogenic at first, chronic M1 macrophage activation actually induces hepatocarcinogenesis. This principle works as follows. Upon initial liver injury, pro-inflammatory and anti-carcinogenic M1 KCs are activated by stimuli such as lipopolysaccharides (LPS), IL-12, interferon (IFN)-γ, TNF-α and granulocyte-macrophage colony-stimulating factor (GM-CSF) [74]. M1 KCs subsequently secrete a number of pro-inflammatory cytokines, including TNF-α, IL-1β, IL-6, IL-12, CCL2 and CCL5, and increased amounts of ROS and nitric oxide synthase (NOS) [68,74,75,76,77]. This yields a pro-inflammatory, anti-carcinogenic environment, characterized by a high IL-12, high IL-23, high NOS and low IL-10 phenotype [74]. In the context of NAFLD-NASH, the fatty acid palmitic acid (PA) induces M1 polarization through hypoxia-inducible factor (HIF)-1α [78]. The anti-carcinogenic properties of these macrophages consist of the trapping, phagocytosing and lysing of tumor cells [79]. Moreover, an enhanced tumor-antigen presenting ability of M1-type macrophages promotes cytotoxic functions by cytotoxic (CD8+) T cells and NK cells, which promote tumor cell apoptosis. Furthermore, M1 macrophages can trigger a T helper (Th)1 immune response [80] and exhibit strong anti-carcinogenic activity by the production of ROS and NOS, which, when expressed in high amounts, prompts autophagy and apoptosis of cancer cells [81,82,83]. Yet, side effects of chronic M1 macrophage activation actually induce hepatocarcinogenesis. These side effects include; hepatocyte (oxidative) DNA damage [38,84], the induction of damage-prone DNA-repair responses and dysplasia, hepatocyte proliferation by activation of intracellular signaling pathways and impaired tumor cell death through flawed apoptosis [18,32,85]. As such, chronic M1 macrophage activation actually promotes hepatocarcinogenesis. 

### 3.2. Pro-Carcinogenic M2-Type Macrophages 

When KCs are exposed to IL-4, IL-10, IL-13, IL-33, glucocorticoids or Toll-like receptor (TLR) ligands, they differentiate into anti-inflammatory and pro-carcinogenic M2 macrophages [74,76,80,83]. Moreover, in the setting of NAFLD-NASH, the unsaturated fatty acid oleic acid (OA) promotes M2 macrophage polarization [78]. M2 macrophages exhibit high phagocytic capacity and produce high levels of IL-4, IL-10, IL-12 and transforming growth factor (TGF)-β [75,76,77]. M2 macrophages promote a Th2 immune response and promote angiogenesis, tissue remodeling and repair [80,86]. Apart from the classically activated M1 phenotype and the alternatively activated M2 phenotype, the existence of more specific M2 subtypes underscores the diversity of macrophages [76], with the M1 and M2 phenotypes representing either end of the spectrum [76,87]. 

### 3.3. Switch from M1- to M2-Type Macrophages Stimulates Progression of NAFLD-Related HCC

The switch from M1- to M2-type macrophages plays an important role in the initiation of carcinogenesis and the progression of NAFLD-related HCC [64,88,89]. Several mechanisms facilitate transition from M1 to M2 macrophages. The upregulation of peroxisome proliferator-activated receptor (PPAR)-γ shifts lipid-induced macrophage polarization from M1- to M2-phenotype by interacting with NF-κB signaling induced by IL-4 and IL-13 [78,90]. NF-κB and STAT1 expression, on the other hand, shift macrophage polarization towards the M1-phenotype via the PI3K/Akt signaling pathway [91]. Insulin suppresses NF-κB and STAT1 expression while promoting PPAR-γ signaling and thus promoting M2 polarization [78,91,92]. Janus kinase (JAK)/STAT6 signaling, which is induced by IL-4, and IL-6/STAT3 and JAK3/STAT3 signaling are also involved in the polarization from M1 to M2 macrophages [90,91,93]. Moreover, TGF-β has been shown to promote M2 polarization [94,95] and IL-10, secreted by M2 macrophages has the ability to promote apoptosis of M1 macrophages, thus further tipping the balance towards an M2 phenotype [74,96]. These mechanisms, however, are not unique to NAFLD and can be seen across various chronic liver diseases. Additionally, numerous other factors, such as microRNAs, Notch signaling pathway [91], Nogo-B expression [97] and wingless-related integration sight (Wnt)/β-catenin signaling [98] are involved in M1 to M2 macrophage switching. The mechanisms that support macrophage polarization are manifold and the induction of polarization probably depends on the co-regulation of multiple signaling pathways [91]. Figure 1 shows a schematic summary of the opposing roles of M1- and M2-type macrophages and the mechanisms underlying polarization and phenotype switching in the context of NAFLD. There is extensive heterogeneity among the macrophage populations and a high proportion of macrophages share both M1- and M2-phenotype characteristics [77,83,89,99,100]. This underscores the complexity and heterogeneity of these processes and suggests the presence of additional pathways. 

### 3.4. Tumor-Associated Macrophages (TAMs)

Once HCC develops, the tumor microenvironment (TME) exacerbates the polarization of macrophages towards the M2-phenotype [68]. The TME, consisting of cancer-associated fibroblasts (CAFs), HSCs, endothelial cells, and immune cells, secretes CCL2 and macrophage colony-stimulating factor (M-CSF) which stimulates circulating monocytes to migrate into the liver and subsequently differentiate into tumor-associated macrophages (TAMs) [80,83]. TAMs are mainly polarized towards the M2-phenotype and secrete high levels of IL-10, whilst expressing low levels of pro-inflammatory cytokines, ROS and nitric oxide. Moreover, TAMs are poor antigen-presenting cells and drive tumor growth via the suppression of an effective cytotoxic response [80,83]. Recently, triggering receptor expressed on myeloid cells 2 (TREM2) has been identified as a specific marker of TAMs in different human cancer models including HCC [101]. Enrichment of TREM2+ TAMs is associated with poor clinical outcome in patients with HCC [102] and knock-out of TREM2 was found to suppress the growth of HCC in an in vivo murine model [103]. This is in line with the immunosuppressive role often attributed to TREM2+ TAMs [101]. On the other hand, several studies also point to a tumor suppressive role of TREM2+ TAMs in the context of hepatocarcinogenesis [104,105]. More research is warranted to elucidate the exact role TREM2+ TAMs play in hepatocarcinogenesis [101,103]. 

The location of TAMs within the tumor itself is indicative of their function. Soluble mediators secreted by tumor cells can trigger the early activation of monocytes in the peritumoral stroma whilst inducing immunosuppressive macrophages in the cancer nests [80,106]. Macrophages in the peritumoral stroma produce significant levels of pro-inflammatory cytokines. Of these, IL-1β, IL-6 and IL-23 promote Th17 cell expansion, whereas TNF-α and IL-10 promote the autocrine upregulation of programmed death ligand (PD-L)1 on the surface of those cells [107,108]. As such, tumor cells are able to reeducate macrophages; upon initial exposure to the TME, macrophages are driven towards a pro-inflammatory and anti-carcinogenic M1-phenotype, whilst macrophages in close proximity to tumor cells are driven towards an immunosuppressive phenotype, thus failing to trigger an effective antitumor immune response [80,106,109].

## 4. Fibrogenesis

KC activation and liver injury drive activation of HSCs and subsequent fibrogenesis. In a healthy liver, HSCs are localized in the subendothelial space of Disse, representing ~8–10% of all resident liver cells [110]. The physiological roles of quiescent HSCs include storage of vitamin A, synthesis of ECM and matrix-degrading metalloproteinases, and the regulation of sinusoidal blood flow [111,112,113,114]. Many factors contribute to HSC activation in the context of NASH. Firstly, hepatocellular damage, either caused by metabolic stress, oxidative stress, inflammatory stimuli and qualitative and quantitative changes in ECM, are important drivers of HSC activation [115]. Secondly, the pro-inflammatory environment triggers the activation of HSCs. This is mediated by several pro-inflammatory cytokines, specifically platelet-derived growth factor (PDGF), TGF-β, TNF-α, IL-1 and several chemokines which originate mainly from activated KCs [49,77]. Single-cell RNA sequencing together with genetic ablation of subpopulation-enriched mediators revealed a dual role of specific subpopulations of HSCs [116]. Quiescent or weakly activated HSCs, characterized by the production of cytokines and growth factors including hepatocyte growth factor (HGF) exert a protective function against hepatocyte death and HCC development [116]. Yet, as liver injury continues, a dynamic shift between tumor-suppressive and tumor-promoting HSC subpopulations occurs, the latter of which is characterized by a highly active, myofibroblastic phenotype and leads to increased deposition of type I collagen and other matrix proteins, thus increasing matrix stiffness [116]. Moreover, these myofibroblastic HSCs promote the proliferation of hepatocytes and secrete increased amounts of pro-inflammatory as well as profibrogenic cytokines [110,116]. These profibrogenic cytokines/growth factors include TGF-β [110,117,118,119] and PDGF [110], two potent profibrogenic cytokines, whose production is not limited to HSCs and are also produced by macrophages [120,121]. Furthermore, activated TGF-β enhances the autocrine expression of TGF-β and prolongs the survival of activated HSCs by reducing apoptosis [122]. TNF-α and IL-1 stimulate the activation of HSCs [49] and pro-inflammatory cytokines produced by macrophages, including TNF-α and IL-1β, maintain HSC survival through the NF-κB pathway [77]. Integrins, which are cell surface receptors that act as mechanoreceptors by relaying the information from cell to cell, and from the ECM to cells, and vice versa, can interfere with TGF-β1 and PDGF and can modulate the proliferation and survival of hepatocytes and HSC [123]. This process relies on a variety of signaling pathways, among others, Hh signaling and MAPK and ERK signaling [49,124,125,126,127,128].

### 4.1. Matrix Stiffness Stimulates NAFLD-Related Hepatocarcinogenesis 

Excessive collagen deposition increases the stiffness of the ECM, which in turn promotes the deposition of additional collagen by HSCs [129]. ECM stiffness can stimulate hepatocarcinogenesis through the activation of the Hippo-Yes-associated protein (YAP)/transcriptional coactivator with PDZ-binding motif (TAZ) signaling pathway [115,129]. Once HCC has been established, matrix stiffness modulates HCC proliferation through a wide variety of signaling pathways [49,130]. ECM turnover is regulated by matrix metalloproteinases (MMPs), which are capable of degrading components of the ECM, and the tissue inhibitor of metalloproteinases (TIMPs), which acts as the endogenous inhibitor of MMPs [131,132]. HSCs are the main source of both MMPs and TIMPs, and their secretion is controlled by TGF-β1 and TNF-α [131]. MMPs and TIMPs are implicated in hepatocarcinogenesis [49,131]. TIMP-1, for instance, inhibits tumor apoptosis via stromal-derived factor (SDF)-1/PI3K/AKT signaling [133], and MMP-1 production enhances proliferation, invasion and fibrosis in NASH [49]. Moreover, increased matrix stiffness drives STAT3 activation [131]. These data underscore the role of MMPs and TIMPs in hepatocarcinogenesis [49]. Carcinoma-associated fibroblasts (CAFs), which are mainly derived from HSCs [134], play an important role in tumor growth and metastasis in HCC [135,136,137]. A number of oncogenic factors, including vascular-endothelial growth factor (VEGF), osteopontin (OPN), and TGF-β facilitate cancer development [49]. The positive feedforward loop in tumor cells and CAFs, in combination with oncogenic growth factors, exacerbates HCC progression [138]. 

### 4.2. Liver Sinusoidal Endothelial Cells (LSECs)

Liver sinusoidal endothelial cells (LSECs) form the main component of the liver endothelium and also play a key role in NAFLD-related hepatocarcinogenesis. LSECs are highly specialized endothelial cells situated between blood derived from the portal system, on the one side, and hepatocytes and HSCs, on the other [29]. During early stages of NAFLD, LSECs exhibit a downregulation of pro-inflammatory chemokines, including CCL2 through a MAPK-dependent pathway [139]. This may represent a compensatory mechanism to help prevent disease progression [139]. However, as NASH progresses, LSECs acquire a pro-inflammatory phenotype, characterized by the production of TNF-α, IL-1, IL-6 and CCL2 [29]. Thus, dysfunctional LSECs contribute to the inflammatory response and the activation of KCs, instead of maintaining KC quiescence [140]. Increased cytokines in the portal circulation, including TNF-α and IL-6, and increased intestinal permeability and consequently elevated LPS concentrations in the portal circulation contribute to the switch towards a pro-inflammatory phenotype [141,142,143]. Moreover, lipotoxicity stimulates ROS formation by LSECs, adding to the pro-carcinogenic environment [29,144]. Additionally, LSECs also stimulate fibrogenesis by overexpressing vascular adhesion protein (VAP)-1 in the context of NASH, which in turn leads to HSC activation [145], and increases the pro-inflammatory environment [29]. Moreover, capillarization, which is the loss of LSEC fenestrae, and LSEC dysfunction, characterized by the inability of LSECs to generate vasodilator agents in response to increased shear stress, both promote fibrogenesis [29]. Hh molecules released from ballooned hepatocytes in the context of NASH, lead to the activation of quiescent HSCs and the activation and capillarization of LSECs [146,147]. Activated HSCs and LSECs, in turn, secrete large volumes of Hh molecules, thus creating an autocrine and paracrine positive-feedback loop [148]. Eventually, this increases fibrogenesis in the liver. LSECs can also directly contribute to hepatocarcinogenesis via the expression of fatty acid-binding protein (FABP)4, which directly induces hepatocyte proliferation [149]. All in all, LSECs contribute to the initiation of NAFLD-related HCC, either directly or indirectly, by creating an environment that is susceptible for HCC development.

Figure 2 shows a schematic overview of the role of macrophages, LSECs and HSCs in the process of fibrogenesis and its role in hepatocarcinogenesis in the context of NAFLD. 

### 4.3. Angiogenesis

Pathogenic angiogenesis increases with NASH [29,150,151]. This is mediated in a variety of ways. First of all, chronic inflammation promotes angiogenesis. In this regard, chronic inflammation and fibrosis sustain tissue hypoxia and, thus, induce the release of HIF-1α, a potent pro-angiogenic factor [151]. In addition, pro-inflammatory mediators, including TNF-α and IL-6, prompt a direct pro-angiogenic effect by inducing HIF-1α and VEGF [151], and cytokines and ROS released during NASH can activate the MAPK/ERK pathway, which is involved in cell migration and angiogenesis [29,151]. Activation of both macrophages and HSCs can increase angiogenesis. The activation of macrophages leads to the release of various factors that can induce angiogenesis, inducing ROS, NOS, TNF-α, HIF-1 and VEGF [152,153,154]. The activation of HSCs and the subsequent release of factors such as HGF, PDGF, VEGF and NO, also induces angiogenesis [151,155,156]. Moreover, in the context of NAFLD, the accumulation of FFAs causes damage to and swelling of hepatocytes. This swelling reduces sinusoidal perfusion thus adding to the pro-angiogenic response in the liver [150,151]. Moreover, the increased cellular stress caused by exposure to excessive amounts of FFAs and gut-derived LPS, prompts hepatocytes to release microvesicles containing pro-angiogenic factors [157]. 

LSECs also play a role in angiogenesis in NAFLD-NASH, as they are the main producers of angiopoetin-2, another driver of angiogenesis in inflammatory conditions [158,159]. Inflammation stimulates angiogenesis that in turn worsens inflammation, thus creating a self-propelling cycle. This is underscored by the anti-inflammatory effect of anti-angiogenic agents [29]. Moreover, tumor cells secrete a number of angiogenic growth factors, including VEGF, PDGF, placental growth factor (PlGF) and TGF-β1, to fulfill their need for high oxygen and nutrient supply [151]. However, these new vessels are marked by a disorganized vasculature, and consist of leaking, hemorrhagic and torturous vessels, which leads to poor oxygenation and free passage for tumor cells into the blood circulation, and hence, facilitate metastasis [151]. 

## 5. Immune Escape of Tumor Cells

Macrophages and HSCs play an important role in tumor immune escape [160]. Both cell type are able to inhibit an effective cytotoxic T-cell response by expressing high levels of PD-L1 on their cell surface [77,107,160,161,162,163]. Additionally, HSCs promote T-cell apoptosis and inhibit T-cell proliferation through the C3 pathway [164], and both macrophages and HSCs are able to induce the expansion of myeloid-derived suppressor cells (MDSCs) [165] and Th17 cells [107,160]. MDSCs promote hepatocarcinogenesis in a variety of ways including the recruitment of regulatory T cells via the secretion of IL-10 and TGF-β and inhibition of cytotoxic T cells by stimulating the expression of PD-L1 on tumor cells, thus mediating immune evasion [166,167]. MDSCs also stimulate angiogenesis via the secretion of VEGF [168]. Th17 cell expansion, and its subsequent secretion of IL-17, IL-22 and IL-23, promotes NAFLD-related hepatocarcinogenesis [108,169]. Differentiation towards the Th17 phenotype is driven by multiple cytokines, including IL-6, TGF-β, IL-21 and IL-23, and IL-1β and TNF-α to a lesser extent [170]. Autocrine IL-17 stimulates the expression of PD-L1 on the surface of peritumoral macrophages and HSCs [171]. Apart from aiding immune escape, IL-17 stimulates the expression of IL-1β, IL-10 and TNF-α on macrophages [172,173], and directly induces type I collagen production by HSCs via activation of the STAT3 signaling pathway [174]. Additionally, Th17 cells are able to aggravate the influx of FFA in the liver and can, thus, exacerbate steatosis and induce additional DNA damage in hepatocytes [172]. The secretion of IL-22 is pro-carcinogenic as it stimulates the STAT3 signaling pathway in hepatocytes and, apart from Th17 cells, is also secreted by Th22 cells [170,175,176].

### 5.1. Role of the Adaptive Immune System

CD8+ T cells and CD4+ T cells are the main players in tumor immune escape in the context of NAFLD-related HCC. Tumor immune escape occurs in HCC with different etiologies, but some NAFLD-related specific methods are at play. In the context of NASH, macrophages actively participate in the recruitment and activation of CD8+ T cells and NKT cells. This is mediated by the release of ROS and cytokines [177]. Dendritic cells (DCs) also aid in the activation of CD8+ T cells by the secretion of cytokines, including IL-1β and TNF, and by acting as antigen-presenting cells (APCs) for CD8+ T cells [173,177,178]. Interestingly, in early stages of NAFLD, lipid-rich hepatic DCs are immunogenic, thereby activating T cells, NK cells and NKT cells, whereas lipid-poor DCs are more tolerogenic in nature and are able to induce Tregs [178,179]. 

### 5.2. CD8+ T Cells

CD8+ T cells have important cytotoxic effector functions and as such are responsible for killing cancerous or virally infected cells. CD8+ T cells are the main subset of tumor-infiltrating lymphocytes (TILs) and perform important anti-carcinogenic functions as they are able to kill target cells by the method of lying, which is partly regulated by IFN-γ [69,180,181]. Activated T cells produce HCC-inducing lymphotoxin-α and -β and other mitogenic cytokines, thus aiding hepatocarcinogenesis [182]. This can be seen in the light of tissue damage resulting from chronic inflammation and is consistent with that in other chronic inflammatory diseases [69]. As such, the antibody-mediated depletion of CD8+ T cells as well as pharmacological inhibition of the lymphotoxin-β receptor markedly delayed tumor development in mice with chronic liver injury [183]. Additionally, CD8+ T cells and NKT cells are capable of activating NF-κB and STAT3 signaling in hepatocytes, thus promoting hepatocarcinogenesis [183,184,185,186].

On the other hand, CD8+ T cells perform distinctive antitumor effector functions [180], as previously mentioned. In line with this view, large numbers of CD8+ T cells in human HCC correlate with improved overall survival, longer relapse-free survival and diminished disease progression [69,180,181]. Cytotoxic reactions are mainly mediated by IFN-γ secreting CD8+ T cells and a subset of NK cells [181]. However, in the setting of HCC, the anti-carcinogenic functions of CD8+ T cells and NK cells are impaired [69,180]. This phenomenon is called T-cell and NK-cell exhaustion, respectively. This has been demonstrated by an impaired ability of CD8+ T cells to secrete IFN-γ, thus hampering the lysing of tumor cells in human HCC with different etiologies, including NAFLD [180]. Moreover, chronic inflammation in the context of NASH is accompanied by the accumulation of liver-resident IgA-producing cells, which express PD-L1 and IL-10 [187]. As previously mentioned, this directly suppresses the function of CD8+ T cells, thus partially explaining the occurrence of T-cell exhaustion. 

Regulatory B cells have been reported to promote HCC growth and invasiveness by directly interacting with hepatocytes through the CD40/CD154 signaling pathway [188]. In other respects, the role of the B cell in the development and progression of NAFLD-related HCC remains unclear [189]. T-cell exhaustion has also been linked to the function of neutrophils, which are able to generate neutrophil extracellular traps (NETs). These NETs enhance PD-L1 signaling and thus promote an immunosuppressive environment [190,191]. Interestingly, elevated FFAs stimulate the infiltration of neutrophils into the liver and stimulate the formation of NETs [190]. All in all, it can be stated that CD8+ T cells both cause liver damage and are needed for surveillance of pre-malignant and malignant cells in the liver [69]. Yet, during advanced stages of disease, TILs express markers of exhaustion, including PD-L1, and show increased IL-10 and TGF-β secretion [69]. Moreover, the recruitment of Tregs further diminishes the effectiveness of T cell-mediated tumor surveillance. Tumor-infiltrating Tregs mainly secrete IL-10 and TGF-β, both of which are key inhibitors of CD8+ T-cell function [69,192]. Tregs, in turn, are recruited into the TME by tumor-associated neutrophils and macrophages, partly by KC-derived IL-10 [193]. As such, an increased abundance of Tregs is correlated with poor survival and high recurrence rates [194]. 

### 5.3. CD4+ T Cells

While the number of Tregs increases in NASH, the number of CD4+ T cells decreases [177]. The interaction between neutrophils and naïve CD4+ T cells drives differentiation into Tregs [191]. In the setting of NAFLD, lipid metabolism dysregulation, the release of specific metabolites, including linoleic acids, and increased ROS induce a selective loss of CD4+ T cells [195]. This selective loss of CD4+ T cells accelerates hepatocarcinogenesis in NASH livers [195]. Although the total amount of CD4+ T cells is decreased in the setting of NASH and NAFLD-related HCC, specific subgroups of Th cells are increased [177]. As previously stated, Th17 cells are upregulated, aiding immune escape and aggravating the influx of FFAs and DNA damage [69,171,172]. Th1 cells secrete IL-2, TNF and IFN-γ, the latter of which stimulates CD8+ T cells and has anti-carcinogenic properties [69,170,191]. Th1 drives macrophages polarization towards the pro-inflammatory, anti-carcinogenic M1-phenotype. In early stages of disease, this (together with Th17 activity) represents an important mechanism supporting lobular inflammation during NASH evolution [173]. However, as disease progresses, the differentiation of naïve CD4+ T cells to Th1 cells, becomes inhibited [177]. The role of Th2 cells in the development of NAFLD-related HCC still remains unclear [177].

NK cells exhibit various anti-carcinogenic functions, including anti-fibrotic properties [196]. NK function, however, is diminished during HCC development by STAT3-mediated upregulation of IL-10 and TGF-β, thus inducing NK exhaustion [69,197,198]. 

### 5.4. Complex Interplay between the Innate and Adaptive Immune System

Increased amounts of NKT cells are found in human NASH, cirrhosis and HCC, but it remains unclear whether this propels or inhibits hepatocarcinogenesis [69,185]. NKT cells have both pro- and anti-inflammatory effector capacities through the secretion of IFN-γ, on the one hand, and IL-17 and LIGHT, on the other [173]. LIGHT, also known as tumor necrosis factor superfamily member (TNFSF)14, promotes steatosis by encouraging the uptake of lipids by hepatocytes [185]. Apart from NKT cells, it is also secreted by CD8+ T cells in early stages of disease [173]. Upregulation of LIGHT is seen in the liver of patients with NASH, but not in patients with ASH or chronic HCV infection, and may represent an NAFLD-specific way of disease progression [185]. Moreover, LIGHT has also been demonstrated to block TNF-α induced hepatocyte apoptosis, and stimulates NF-κB signaling in hepatocytes, thus stimulating proliferation [199]. In murine models, LIGHT deficiency prevented NASH and HCC development, and led to a strong reduction of NKT cells in the liver [185]. NKT cells also activate HSCs and thus promote fibrogenesis [200], adding to their pro-carcinogenic properties. 

KCs influence tumor immune escape because of their tolerogenic nature. As such, KCs function as incomplete antigen-presenting cells (APCs) and can even inhibit DC-induced T-cell activation in murine models [201]. Moreover, in advanced HCC, the functions of DCs are impaired, thus preventing the proper activation of antigen-specific T cells [202]. 

Additionally, endothelial cells within HCC can also alter the tumor-associated immune response by promoting T-cell tolerance towards cancer-associated antigens and creating an immunosuppressive environment [203]. This is regulated by the uptake of tumor-cell apoptotic fragments by LSECs and the subsequent cross-presentation of this apoptotic cell material by LSECs to CD8+ T cells, which leads to the induction of CD8+ T-cell tolerance [203]. 

Figure 3 shows a schematic overview of the different cell types involved in and the processes resulting in T-cell exhaustion in the context of NAFLD-related HCC. 

### 5.5. Immunotherapy for NAFLD-Related HCC

A growing body of evidence highlights the difference in the effects of immunotherapy for HCC in the context of NAFLD-NASH to those of other etiologies [177,204,205]. In a mouse model of NAFLD-related HCC, immunotherapy with anti-PD1 led to an increase in the incidence of NAFLD-related HCC and in the number and size of tumor nodules. Interestingly, this increase was correlated with increased hepatic CD8+ programmed death (PD)+ T cells and TNF+ T cells and was prevented by depletion of CD8+ T cells or TNF neutralization. This suggests an important role of CD8+ T cells in the induction of NAFLD-related HCC [205]. This is in contrast to other mouse models with HCC from other, non-NAFLD, etiologies, where PD1 immunotherapy induces tumor regression [206]. Impaired tumor surveillance and enhanced T cell-mediated tissue damage are suspected to underlie this difference, adding to the risk of transition to HCC in mice with NASH [204,205]. These results are supported by a meta-analyses of three large randomized controlled phase III trials of immunotherapy in patients with advanced HCC [205]. While benefit is evident regardless of etiology, the meta-analysis suggests that patients with NAFLD etiology may benefit less after anti-PD1 treatment. A combined therapy using both anti-PD-L1 and anti-VEGF has been shown to prolong survival [207,208,209] and is now the recommended first-line medical therapy regardless of etiology, but further investigations are warranted to elucidate the underlying mechanisms relevant to variability in response rates [177,210]. 

## 6. Conclusions

In this review, we identified the key roles of non-parenchymal cells in both initiation and progression of NAFLD-related HCC. Upon activation, macrophages prompt an effective pro-inflammatory, anti-carcinogenic response. However, as inflammation becomes chronic, this pro-inflammatory response actually drives genomic instability via the secretion of ROS and hepatocyte proliferation, amongst others. Moreover, as inflammation continues, the pro-inflammatory, anti-carcinogenic M1 macrophage response shifts towards an anti-inflammatory, pro-carcinogenic M2 macrophage response. This further drives error-prone DNA repair and increases genomic instability and the chance of HCC initiation. Moreover, M2 macrophages suppress an effective cytotoxic response by secreting anti-inflammatory cytokines, such as IL-10, expression of PD-L1, and by acting as incomplete antigen-presenting cells, thus aiding immune escape. This is exacerbated by a combined effect of neutrophils, plasma cells, LSECs, MDSCs and regulatory T cells that drive CD8+ T-cell exhaustion, characterized by a IFN-γ^low^, IL-10^high^ and TGF-β^high^ phenotype. 

The activation of HSCs by macrophages prompts their differentiation into myofibroblasts, which in turn secrete additional ECM. The increase in ECM stiffness contributes to hepatocarcinogenesis and promotes hepatocyte proliferation by modulating various signaling cascades. LSECs can also induce HSC activation and induce hepatocyte proliferation. This mechanism plays an important role in angiogenesis and increases the potential of metastasis. 

The process of HCC initiation and subsequent progression is highly complex and depends on the underlying etiology. The differences in pathogenesis between NAFLD-related HCC and HCC induced by, for example, viral-induced HCC, warrants further research to elucidate the differences in treatment response. In this regard, impaired immune surveillance in NAFLD-related HCC may partially explain the difference in response to immune checkpoint inhibitors compared to viral-induced HCC. Specific molecular mechanisms leading to HCC may serve as potential new target agents. As such, therapies combating T-cell exhaustion, such as anti-PD1 treatment, could become viable treatment options for patients with NAFLD-related HCC. 

## Figures and Tables

**Figure 1 cancers-15-01308-f001:**
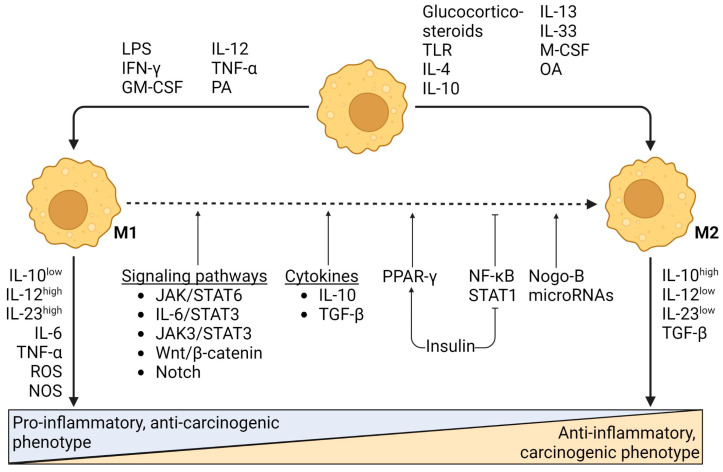
Schematic summary of M1- and M2-type macrophages and the mechanisms underlying polarization and phenotype switching in the context of NAFLD.

**Figure 2 cancers-15-01308-f002:**
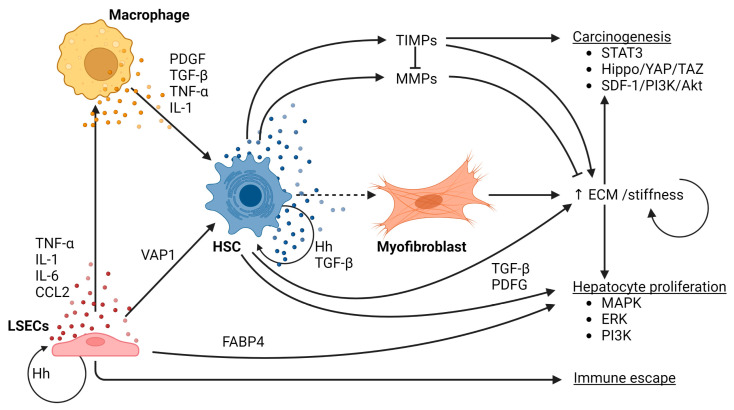
Schematic overview of the role of macrophages, LSECs and HSCs in the process of fibrogenesis and its role in hepatocarcinogenesis in the context of NAFLD.

**Figure 3 cancers-15-01308-f003:**
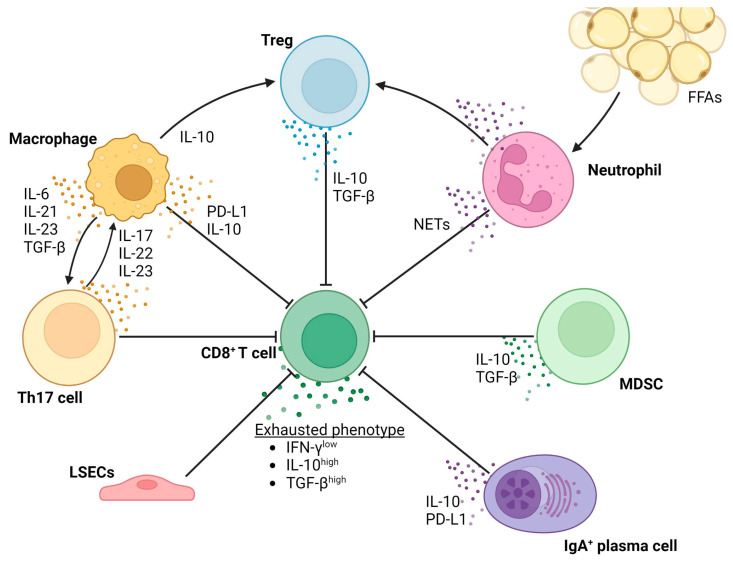
Overview of the processes resulting in T-cell exhaustion in the context of NAFLD-related HCC.

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
