# Peer review of "Non-Parenchymal Cells and the Extracellular Matrix in Hepatocellular Carcinoma in Non-Alcoholic Fatty Liver Disease"

_cancers, 2023, doi:10.3390/cancers15041308_

Round 1

Reviewer 1 Report

Peer review cancers-2149815

In this review, van Son et.al summarize the work that has been done on the role of non-parenchymal cells in the initiation and progression of NASH induced HCC. In general, the manuscript is well written, however I would recommend some adjustments before publication.

 Minor comment:

1.     In the abstract LSEC are referred to as sinusoid endothelial cells as well as sinusoidal endothelial cells. Since only the latter is used further in the paper, I recommend adapting sinusoid endothelial cells to sinusoidal.

2.     Line 180 in the paragraph of the Macrophage types, is written that M1 macrophages are mainly pro-cancer growth, while they induce cancer cell autophagy and apoptosis. I guess then they have anti-cancer activity.

Major comments:

1.       When reviewing non parenchymal cells in HCC most studies will be on the immune compartment of the TMA. Now, in the paper macrophages are separated from the other immune cells T and B cells by the paragraphs on HSC and LSEC. I suggest to first discuss the HSC and LSEC and then the paragraphs on macrophages and other immune cells.

2.       In the section on fibrogenesis and stellate cells/CAFs most papers referred to in this review are other older reviews. More recent research papers should be included, e.g. the studies by the group of Prof. Schwabe where clear subpopulations of CAFs in liver cancer (incl. NASH-HCC) have been identified using scRNA-SEQ. Furthermore, in these studies col1 was deleted in CAFs and the effect thereof on HCC development was addressed. 

3.       In the last part of the fibrogenesis paragraph, some pathways are listed. It is not clear why these were selected and not others. This should be either clarified or extended.

Author Response

Dearest reviewer. Thank you for reading our manuscript and providing us with valuable comments on how to improve on our manuscript. Please find the response to the comments and the actions taken to ameliorate the manuscript below.  

Minor comments:

  1. Thank you for point out this inconsistency. We have altered ‘sinusoid endothelial cells’ to ‘sinusoidal endothelial cells’ according to your recommendation.

  1. Again, thank you for pointing out this mistake. Indeed, we meant to discuss the anti-carcinogenic effect of M1 macrophages. The manuscript has been adjusted.

Major comments:

  1. Thank you for your valued suggestion. Following the review process and subsequent adaptations to the manuscript, we have expanded paragraph on the process of fibrogenesis on the development of NAFLD-related HCC. As part of this adjustment, we have put more emphasis on the role of macrophages in fibrogenesis. As such, the distinction between macrophages and HSCs in the different paragraphs is less apparent. Therefore, we have decided to maintain the order of the paragraphs.

  1. Thank you for this highly priced suggestion. We have added additional information on more recently published studies, including the one by Prof. Schwabe on CAFs in liver cancer. Moreover, we have added information about the different subtypes of HSCs which are presented in those more recent studies.

  1. We have put more emphasize on the large variety of pathways at play in this process and have made it more clear that these pathways are discussed here as an example. Thank you for the suggestion.

Reviewer 2 Report

 The manuscript by van Son et al. (Cancers 2149815) has been designed in order to review the role of hepatic non-parenchymal cells in the pathogenesis of NAFLD/NASH – related HCC, with a specific focus on the role on both innate and adaptive immune response as well as on critical processes such as fibrogenesis and angiogenesis.   

The review by van Son et al. is reasonably up-to-date and comprehensive, offering an appreciable overview of the state of the art on the selected topic. I can offer the following comments:

1. Authors have correctly introduced data and concepts concerning the conventional view of the opposite role of M1 vs M2 macrophages in NAFLD/NASH progression as well as, of course, the role of TAMs in HCC. I would ask Authors to introduce at least a short paragraph concerning the emerging evidence for a role of TREM2 and CD9 positive macrophages  at least in NAFLD/NASH progression (data on their role in liver carcinogenesis are still essentially missing).

2. Authors have of course described the role of HSC and of MFs in fibrogenesis. I would suggest to mention also the data remarking a pro-angiogenic role of these cells.

Author Response

Dearest reviewer. Thank you for reading our manuscript and providing us with valuable comments on how to improve on our manuscript. Please find the response to the comments and the actions taken to ameliorate the manuscript below.  

  1. Thank you for your valuable suggestion. We have added a section on the possible role of TREM2+ TAMs in the progression of NAFLD-related HCC. We believe this extra piece of information adds on the relevance of TAMs in NAFLD-related HCC and provides the manuscript with data on recent discoveries in this field. We have also looked into the role of CD9+ macropahges in the development of NAFLD-related HCC but, unfortunately, the evidence of the role of these cells in liver disease is, up to our knowledge, limited to steatosis. As such, we have omited them from further discussion as our manuscript is focussed on the development of HCC. 

  1. Thank you for your comment. In the paragraph dedicated to angiogenesis, we have elucidated several pathways in which macrophages and HSCs drive angiogenesis in the context of NAFLD-related HCC.

Round 2

Reviewer 1 Report

The authors have made the requested changes accordingly. I have no additional comments.